# Integrating the Tumor Microenvironment into Cancer Therapy

**DOI:** 10.3390/cancers12061677

**Published:** 2020-06-24

**Authors:** Sabina Sanegre, Federico Lucantoni, Rebeca Burgos-Panadero, Luis de La Cruz-Merino, Rosa Noguera, Tomás Álvaro Naranjo

**Affiliations:** 1Department of Pathology, Medical School, University of Valencia—INCLIVA Biomedical Health Research Institute, 46010 Valencia, Spain; ssanegre@incliva.es (S.S.); lucantoni.federico@hotmail.it (F.L.); reburpa@alumni.uv.es (R.B.-P.); 2Low Prevalence Tumors, Centro de investigación biomédica en red de cáncer (CIBERONC), Instituto de Salud Carlos III, 28029 Madrid, Spain; 3Department of Oncology, Hospital Universitario Virgen Macarena, 41009 Seville, Spain; ldelacruzmerino@gmail.com; 4Department of Pathology, Hospital de Tortosa Verge de la Cinta, Catalan Institute of Health, Institut d’Investigació Sanitària Pere Virgili (IISPV), 43500 Tortosa, Spain; 5Department of Morphological Science, Medical School, Rovira i Virgili University, 43201 Reus, Spain

**Keywords:** immune therapy, metabolism, microbiota, biomarker discovery, prognostic tools, vitamin D3, stromal reprogramming, mitochondria, mechanotransduction, metformin

## Abstract

Tumor progression is mediated by reciprocal interaction between tumor cells and their surrounding tumor microenvironment (TME), which among other factors encompasses the extracellular milieu, immune cells, fibroblasts, and the vascular system. However, the complexity of cancer goes beyond the local interaction of tumor cells with their microenvironment. We are on the path to understanding cancer from a systemic viewpoint where the host macroenvironment also plays a crucial role in determining tumor progression. Indeed, growing evidence is emerging on the impact of the gut microbiota, metabolism, biomechanics, and the neuroimmunological axis on cancer. Thus, external factors capable of influencing the entire body system, such as emotional stress, surgery, or psychosocial factors, must be taken into consideration for enhanced management and treatment of cancer patients. In this article, we review prognostic and predictive biomarkers, as well as their potential evaluation and quantitative analysis. Our overarching aim is to open up new fields of study and intervention possibilities, within the framework of an integral vision of cancer as a functional tissue with the capacity to respond to different non-cytotoxic factors, hormonal, immunological, and mechanical forces, and others inducing stroma and tumor reprogramming.

## 1. Introduction

Since the 1980s, cancer research has focused on developing new therapeutic agents targeting DNA alterations and the search for a suitable cure, rather than understanding cancer as an integrated system composed of several modules. To date, the ability of cancer cells to survive for a prolonged time is still incompletely understood. In this context, we are beginning to include the role of stem cells in the tumor evolution process and to point out cellular pathways as a physiological adaptive process like a “wound that never heals” [1]. The idea that cancer originates as a consequence of a malignant cellular development can be considered in a multi-level framework where different processes such as senescence, regeneration, wound healing and proliferation play a key role.

The majority of tumors have a heterogeneous cellular population, and the vision that cancer originates from clonogenic expansion from a single mutated cell could be classed as simplistic and inexact. Microscopic observation of tumors reveals a multicellular 3-dimensional complex tissue underlying developmental alterations and a variable degree of morphological, immunophenotypic, and genomic heterogeneity. Indeed, the strategy used until recently of targeting one gene/protein/process at a time has proved unsuccessful. The Cancer Genome Atlas (TCGA), a public repertoire of genomic, epigenomic, transcriptomic and proteomic data, available to anyone in the research community, that leads to improvements in the ability to diagnose, treat and prevent cancer, represents a remarkable feat [2]; however, every tumor shows several mutations in the genome, and with current knowledge, the search for drug candidates for each mutation seems unfeasible. Indeed, even the most clinically-promising drugs, such as tyrosine kinase inhibitors, represent a small advancement in comparison to the diversity of processes and pathway interactions regulated by these enzymes [3]. The majority of breast, colon, and pancreatic cancers encompass between 50 and 80 mutated genes [2] and thousands of mutations in single cancer cells; in addition, this single tumor might exhibit a remarkable degree of morphological and genetic heterogeneity at different stages of its development.

The tumor microenvironment (TME), which surrounds and interacts with tumor cells including the extracellular matrix (ECM) elements, stromal cells, blood and lymph vessels, nerve fibers and the signaling molecules, has been put forward to provide a more complex overview of how cancers develop and progress [4,5]. Researchers, oncologists, and pathologists must understand that the complexity and heterogeneity found at a cellular and molecular level are influenced by a pro-tumorigenic TME and systemic macroenvironment and vice versa. Thus, in this review, we discuss recent developments in cancer and its ecosystem biomarkers and technologies based on an advanced understanding of the pathophysiological nature of cancer and its environment. We analyze the TME as the modulator of the dynamic ecosystem and examine the intrinsic and extrinsic systems capable of inducing TME reprogramming.

## 2. Is the TME Potentially Reprogrammable?

The concept of cancer as a disease based on tissues and not only on genetic alterations at the cellular level [6] opens the door to a new understanding of cancer and its treatment. Growing evidence shows that the stroma is decisively involved in carcinogenesis [7]. An emerging strategy for cancer treatment is to revert the malignant phenotype by targeting the TME instead of or additionally to the cancer cell population [8], to modify the relationship between the cells and the stromal compartment to obtain a response on the substrate where tumors grow. Breast carcinoma cells injected into the adipose tissue of syngeneic rats not exposed to the carcinogen, reverse their malignant phenotype and acquire benign features [9]. In addition, components of the ECM such as type I collagen, basement membrane components, and the presence of normal fibroblasts show the ability to reverse the tumor phenotype through interactions with a favorable TME [10]. These observations present cancer as a functional tissue with the capacity to respond to different local and distant non-cytotoxic factors, hormonal, immunological, mechanical, and other influences, capable of reprogramming their stroma and cells.

## 3. Emerging Systems of TME Reprogramming 

Several local factors that shape the immune response, the ECM, and the adaptive process of angiogenesis contribute to the TME and tumor evolution. Host factors such as intestinal dysbiosis, neurotransmitters/neurohormones implicated in stress response, host and tumor metabolism, infections, surgical and physico-chemical stimuli can also impact on treatment response, activate the hypothalamic–pituitary–adrenal (HPA) axis and increase the risk of metastasis [11,12].

Tumors cells become aggressive through different mechanisms, such as epithelial-mesenchymal transition (EMT), which provides cancer cells with invasion and motility properties. This suggests the utility of modulating EMT and the inverse process, the mesenchymal to epithelial transition (MET), at a clinical level [13] to revert the malignant phenotype. In fact, this reported EMT-MET cycle implies that changes in cell plasticity during tumor progression are temporary and could be reversed [14]. EMT-MET can lead to the acquisition of stem cell-like features, to modify physico-chemical immunoregulatory properties and genetic, epigenetic, and functional behavior at all levels [15]. EMT is characterized by remodeling of the ECM and factors secreted by the mesenchymal stem cells that actively modulate several oncogenic signaling pathways such as JAK/STAT, Hedgehog, Wnt, Notch, NF- κβ among others, to help stem cells maintain their properties [16]. MET induces regrowth and re-establishment of cancer cells at secondary/metastatic sites. TME factors like Runx2 expression, loss of promoter methylation, and/or miRNAs, can contribute to MET at the metastatic site [16].

Analysis and insight into these processes could support stratifying tumoral heterogeneity at the morphological, immunophenotypic, and genetic level throughout the stroma thus allowing us to identify targetable features [17]. Beyond the dynamic and progressive genetic alterations of cancer cells, other factors show multiple connections on the tumoral tissue with functional response capacities, such as TME molecules, hormones, cytokines, and neurotransmitters, as well as the macroenvironment, where the intestinal microbiota and external factors ranging from stress to medication intake play a role.

### 3.1. Intrinsic Systems Capable of Inducing TME Reprogramming

#### 3.1.1. Remodeling TME to Enhance Antitumor Immune Activity

The immunological landscape of the TME plays a pivotal role in tumor progression. Cancer immunotherapy has gained growing importance in the last decade and several therapeutic strategies are based on the reactivation of the immune system. However, most immunotherapies employed to date are administered systemically, leading to toxicity [18]. Targeting the TME by intratumoral injection of immunomodulators has been studied widely as a method to overcome this limitation, as have combinatorial strategies, stimulatory cytokines administration, inhibitory cytokine blockade and inhibition of immune checkpoints to restore immunological capability at the tumor site [19]. Although immune checkpoint blockade (ICB) appears very effective in a subset of patients, non-responders still remain very high. Several mechanisms for ICB resistance have been proposed [20] but a deeper understanding of the immune landscape at the TME is required for better patient stratification so they can benefit from ICB therapy. Characterization of the immune cell subpopulations is currently assessed using methods such as fluorescence-activated cell sorting (FACS) or immunohistochemistry (IHC)-staining. Novel transcriptome-based cell-type quantification algorithms are being optimized to provide cell-type signatures for immuno-oncology [21]. However, increasing the spatio-temporal resolution of these techniques is still necessary to achieve a more comprehensive view of the immune TME milieu, predict response to ICB and encourage the discovery of new immunotherapeutics [22].

The reciprocal interaction between cancer cells and TME determines the recruitment, activation, and reprogramming of stromal, inflammatory, and immune cells [23] (Figure 1). We have found emerging evidence of the role of ECM remodeling, structural plasticity, and mechanical forces in regulating immune cell trafficking, activation, and immunological synapse formation [24]. 

Cancer cells can downregulate the expression of endothelial adhesion molecules secreted into the TME and required for the transendothelial migration of leucocytes to diminish exposure to cytotoxic effector cells and evade the immunological response [25]. In addition, activation and proliferation of leucocytes have been shown to be a mechanosensitive process relying on substrate stiffness [26,27,28]. Leucocytes are also able to exert a certain degree of mechanical force on the ECM and surface of interacting cells [29].

Several components of the TME have been shown to play a direct role in shaping the immune response. Cancer-associated fibroblast (CAF)-mediated ECM remodeling and fibrosis contribute towards an immunosuppressed and pro-tumorigenic TME by affecting the recruitment and function of various innate and adaptive immune cells [30]. A dense architecture barrier can be imposed by ECM elements such as collagen, hyaluronic acid (HA), and laminins. Whereas high molecular weight HA provides structural integrity and increases regulatory T (Tregs) cell activity to suppress the immune system [31,32], laminins prevent transmigration and polarize leucocytes [33]. In addition, ECM remodeling enzymes such as metalloproteinases and matrikines direct polarization and activation of immune cells, acting as cytokines and chemokines promoting IL expression and T cell chemoattractant [34,35,36].

Given these findings, we suggest adding a systematic study of the relationship between ECM remodeling and the inflamed stromal components of the TME to the current characterization of the immune TME. In an attempt to methodize evaluation of the host immune response in regarding the TME, the Immunoscore assay quantifies immune cell density to predict patient prognosis and suggest clinical treatment [37]. This takes into account tumoral/TME heterogeneity and from a spatial point of view considers the core of the tumor and its invasive margin, with the contribution of T-cell subpopulations (CD3, CD8, and CD45RO); this could be included in the anatomopathological report as a classification in five levels (0–4), giving more information regarding the treatment options for decision making in a personalized medicine approach [38]. Production of IFN-γ induces PD-L1 (programmed death-ligand 1, also termed B7–H1) tumoral expression correlating with the presence of tumor-infiltrating lymphocytes (TIL), which in turn produce IFN-γ (Figure 2a). The presence or absence of PD-L1 and TIL determines the classification into 4 subtypes listed as TIME (Tumor Immune MicroEnvironment): T1 (PD-L1−, TIL−), T2 (PD-L1+, TIL+), T3 (PD-L1−, TIL+) and T4 (PD-L1+, TI−) (Figure 2(b1–b4)) although the existence of the latter is under debate because, in the absence of TIL, PD-L1 is not expected [39]. T2 tumors have been shown to correlate with better response to anti-PD-1 therapy [40]. Although T3 tumors have TIL, they do not express PD-L1, most likely due to a cellular dysfunction where T effector cells are not able to produce IFN-γ [41]. Co-stimulation of T3 tumors with OX-40 or 4-1BB agonists could disrupt the T-cell tolerance [42] and reprogram the TME into a more treatable tumor. However, the majority of tumors are classified as T1 and T4, both lacking TILs possibly owing to an active suppression of inflammatory infiltration or failure of tumor antigen presentation. The TIME that shows abundant immune cells in the periphery, but is empty of cytotoxic lymphocytes (CTL) in the tumor core, is called TIME infiltrated-excluded (I-E). It possesses a high number of CTLs with low expression of activation markers and low CTL infiltration in the tumor core, meaning that adaptive immunity is unable to recognize malignant cells. On the other hand, infiltrated-inflamed (I-I) TIME is characterized by abundant CTLs that express programmed cell death protein 1 (PD-1) with an antitumoral cytotoxic capacity, therefore considered highly immunogenic tumors. Several strategies could induce the TME reprogramming by promoting an inflammatory infiltration, from focal radiation, locally administered oncolytic viruses, and cryotherapy to the use of anti-cytotoxic T-lymphocyte-associated protein–4 antibody, cancer vaccines, and adoptive T-cell therapy. Costimulatory targeting of 4-1BB or OX40 could also potentially increase tumor infiltration in T1 and T4 tumors [42,43]. A pan-cancer combined quantitative analysis of genomics and proteomics of the tumor matrisome (ECM and its related components) not only relates the tumor matrisome index to mutational load and tumor pathology but also predicts survival rates. Worthy of note, high tumor matrisome index tumors revealed enrichment of specific tumor-infiltrating immune cell populations, along with signatures predictive of resistance to ICB, and clinically targetable immune checkpoints [44]. 

#### 3.1.2. The Nervous System (NS), Adrenaline and Glucocorticoids, and Their Role in Metastases 

In cancer both the protective mechanisms of the immune system and the protective influence of the NS are lost [45], leaving the peripheral nervous system (PNS) as a critical part of the tumor stroma, its function, and its structure [46]. While there is increased insight into the function and significance of nervous components of TME, the neurobiology of cancer is an emerging discipline in oncology. It provides growing evidence on the master regulatory effect on immunity of neurotransmitters and psychosocial factors [47] and shows that tumors are not isolated structures, but interact with different systems, directly cell to cell, through electromagnetic signals, as well as through neuronal signaling molecules [48].

TME components such as nerve fibers are important tissue elements in defining the intra- and peritumoral neural milieu [49,50,51]. Clinical and in vitro experiments have shown that tumor innervation release neurotransmitters, neuropeptides, and neurotrophins, acting directly on receptors expressed by cancer cells and modulating signaling, apoptosis, angiogenesis, metastases, and progression [52,53]. In fact, in gastric cancer patients, tumor stage has been correlated with neural density whereas vagotomy reduced the risk of gastric cancer [54]. The tumor cells emit humoral and nervous signals that not only reach the brain, which uses the information to modulate the neuroendocrine and immune system [55] but can spread via perineural invasion of surrounding nerves and related structures. Notably, perineural invasion relates to poor prognosis, correlating with decreased overall and disease-free survival time in colon, pancreatic, gastric, and head and neck carcinoma [56,57,58,59].

Most human tumors express surface adrenergic receptors, which when activated by stress catecholamines play a role in facilitating tumorigenesis and tumor progression [60]. In lung and breast in vitro cancer models, stimulation of β-adrenergic receptors resulted in the increased metastatic potential of cancer cells, among others via natural killer (NK)-cell, macrophage signaling, or osteoblast stimulation [61]. In vitro, in vivo, and clinical studies show that stress-related processes can affect the pathways involved in cancer progression, including immune regulation, angiogenesis, and invasion. It has been shown that chronic use of beta-blocking drugs (antagonizing norepinephrine and adrenaline (Table 1), is associated with lower recurrence and mortality of breast cancer and malignant melanoma and could to decrease prostate cancer risk [62]. Moreover, low-dose glucocorticoids can suppress ovarian cancer progression and metastasis, probably through upregulation of metastasis suppressor microRNAs, but also via modulation of tumor-associated macrophages and myeloid-derived suppressor cells (MDSCs) in the TME [63]. In addition, the sympathetic nervous system (SNS) regulates pathological gene expression in human tumors, leading to DNA damage repair inhibition, oncogene activation, apoptosis, and anoikis suppression [64]. Other neurotransmitters such as endorphins influence tumor proliferation and electrical stimulation of the hypothalamus, increasing the cytotoxicity of NK cells, while pinealectomy affects the course of breast cancer, an effect reversed by the administration of melatonin [48].

Understanding the interaction between cancer cells and NS is becoming indispensable for the development of new targeted therapeutic intervention. 

#### 3.1.3. Intestinal Microbiota as TME Regulator

In cancer, the intestinal microbiota (IMB), the complex and dynamic population of microorganisms that live in the digestive tracts, is of two-fold importance: on the one hand, for its etiopathogenic role [65] and on the other for its effect on cancer treatment efficacy, both through an impact on TME [66]. Several studies link the IMB to the maturation of the immune system, the structure of the TME, metabolism modulation, response to chemical and immunotherapeutic treatment, and most hallmarks of cancer [66,67,68,69]. The existence of a tumor microbiota, found in situ in the TME, could, therefore, have major physiopathological and therapeutic implications [70,71]. All these functions perhaps partly owing to the privileged bidirectional communication between IMB and NS, through the so-called neurenteric axis, HPA regulator, and the accompanying homeostatic equilibrium triangle made up by the endocrine, immune and NS [72].

Importantly, IMB modulates the immunotherapeutic response of anti-PD-1 in patients with melanoma [73] and epithelial tumors [74], showing how IMB regulation is a key factor in the equilibrium between Treg cells, antigen processing/immunoglobulin-secreting cells and the TME structure [75]. On the other hand, dysbiosis develops a pro-inflammatory environment, deregulates the immune response, and diminishes the concentration of chemotherapeutic agents by increasing desmoplasia in the TME [75]. Metagenomic analyses have shown an enrichment of *Fusobacterium nucleatum* in colorectal carcinoma tissue (Table 1). *Fusobacterium nucleatum* causes immunosuppression and recruits tumor-infiltrating immune cells, thus yielding a pro-inflammatory microenvironment, which promotes colorectal neoplasia progression [76]. Antibiotics adversely affect overall and disease-free survival in cancer, regardless of other criteria, due to the destruction of IMB [74], required for chemotherapy to be effective. Overall survival of 20 months without and 11 months with antibiotics (15 and 8 months respectively in lung cancer) show how the IBM governs immune checkpoints and opens up new approaches for a decisive intestinal ecosystem in resistance to inhibitors of immune response control and modification of TME [77]. 

This opens a new field of research and clinical application and puts forward the IMB as an interesting biomarker that determines the efficacy of immunotherapy treatment [78] (Table 1).

#### 3.1.4. Metabolic Regulation and Mitochondrial Dysfunction of Cancer

Otto Warburg pioneered the study of tumor metabolism [114], which established hypoxia and acidosis as characteristics of cancer. This specific metabolic pattern is based on aerobic glycolysis of tumor cells (Warburg effect), which is a necessary source of substrates for uncontrolled tumor cell growth considering that tumor suppressor oncogenes and genes might be carriers of bioenergetic alterations [115]. Metabolic reprogramming is an essential mechanism by which cancer cells switch to different pathways to obtain the energy necessary to survive and proliferate. This metabolic and bioenergetic shift sustains high proliferation rates, as carbon sources are rapidly diverted to produce lipids, nucleic acids, and proteins [116]. This process is also essential to regulate the interaction between cancer and immune cells, as well as to recruit a variety of immune cells [117]. Cancer metabolism has led to a scientific focus on tumor cell reprogramming of glucose consumption [118] to correct the dysfunctional behavior of tumors. Indeed, it is well accepted that aberrant cancer metabolism is linked to treatment resistance [119]. Glutamine levels, which are decreased in the hypoxic core of the tumor, drive histone methylation, and tumor de-differentiation to lead drug resistance [120]. Glutamine also affects the stroma by changing it to a tumor-promoting environment through increased glutamine-induced autophagy in fibroblasts [121]. Lipid metabolism also supports TME reprogramming; indeed, it has been found that Tregs accumulate lipids and combine glycolysis and fatty acid synthesis and oxidation to survive [122]. In addition, recent studies show that the metabolic state of TME, oxygen levels, acidity, and nutrient availability affect T-cell infiltration, survival, and effector function [123,124]. Furthermore, gradients of extracellular metabolites, levels of ischemia, hypoxia, and lactate act as morphogens and increase stromal stiffness. Among other findings, metabolic alterations of the stroma impact tumor heterogeneity, affecting both morpho-immunophenotypic and genetic features [17]; they contribute to identifying sampling error as a cause of lack of correlation between biomarker research and response to immunotherapeutic treatment; shed light on the spatial structure of the TME depending on the Warburg effect [125] and finally show the ability of tumor cells to reprogram their metabolism and survive the harsh conditions of TME [126]. 

Furthermore, mitochondrial lesions not only affect tumor metabolism but also alter apoptosis and present cancer as mitochondrial dysfunction [127]. The cells act as if there is oxygen shortage, even if there is abundance as HIF-1a maintains the expression of normally inactive genes that make the cell immortal, unable to activate programmed cell death and keep its reproduction program activated [128] (Table 1) through the concatenation of hypoxia, lactate levels, malignancy, and metastatic capacity. While the highly glycolytic tumor cell is very aggressive and invasive, cells without mitochondrial DNA cannot form tumors [129], unless they acquire it from adjacent cells, as occurs through mitochondria donation from stromal cells to tumor-deficient tumor cells. Tumor cells recover respiratory capacity and biological aggressiveness when they take up mitochondrial DNA from TME cells [130]. A promising perspective could be based around the ability of the mitochondria to suppress the malignant phenotype [127].

#### 3.1.5. Mechanotransduction and Biotensegrity as Properties of the TME

The mechanical properties of the tumor stroma are determinants of cell biology and clinical behavior [131]. A recurrent tumor feature is the stiffness of the stroma, through which tumors can be detected by palpation or radiological examination. The main mechanical perturbations of the TME are stiffness of the ECM, elevated interstitial fluid pressure, and/or an increase in solid tension caused by tumor growth [132]. These mechanical properties are difficult to study and evaluate with conventional histological techniques, and advances in the field now stem from a new knowledge area called mechanobiology, an interdisciplinary science branch combining biology, physics and engineering. 

The tumor tissue biotensegrity mechanism is based on tension forces on cells and various elements of the ECM, which receive mechanical impact through specifically designed elements [102]. The physical stimuli at the tissue, cellular and molecular level profoundly affect the chemical signals of the tumor cell, capable of perceiving the mechanics of the substrate and transferring this information to the internal signaling pathways by a process known as mechanotransduction [133]. Therefore, matrix stiffness profoundly influences cell morphology and cellular behavior, and vice versa [131]. Intracellular signaling in response to TME mechanics primarily uses the integrin family and affects tumor suppressor genes, oncogenes, and genes for development and differentiation, relating tissue mechanics to carcinogenesis. In addition, to carry out many processes such as invasion and metastasis, cells need to invade the surrounding tissue, break down cell-cell contacts, remodel cell-matrix adhesion sites, and follow a chemoattractive path through the ECM [134]. During all these steps, tumor cells undergo dramatic morphological changes, based on cytoskeleton rearrangement related to changes in mechanical properties [135].

Currently, several strategies based on mechanotransduction are being studied, such as blocking tumor cell interaction through the glycoprotein vitronectin, which reduces the tumor’s invasive capacity [101], or the use of metformin in cancer, which acts on stromal cells by decreasing ECM rigidity and reprogramming tumor cell metabolism towards less aggressive phenotypes [136]. In addition, the increased deposition of fibrillar proteins of the ECM constitutes an independent prognostic factor [137] regarding the integrated tension system that the cell has to maintain its morphology and function. These elements, together with stromal cells and other intrinsic and extrinsic components and stimuli, determine the phenotypic diversity, gene expression, and therapeutic response of tumor cellularity [138,139]. Development of more realistic and complex model systems, along with biophysical instrumentation for study and real-time remodeling of cell-dynamics, raise hopes of future novel therapeutic strategies in oncology [140,141].

### 3.2. Extrinsic Stimuli with the Capacity to Induce TME Reprogramming

#### 3.2.1. Effect of Surgery 

Surgery promotes physical and emotional stress, leading to activation of neuroendocrine mediating mechanisms which in turn affects immunity and the TME, with a prominent role for catecholamines and prostaglandins in this process [142]. This observation would suggest the benefit of moderating catecholamines and prostaglandins levels which show a strong correlation with survival and cancer progression. For this reason, the use of β-blockers, which act by modulating stress levels and catecholamines, should be considered as an additional measure for TME reprogramming [60]. The systemic inflammatory response induced after surgery promotes the appearance of tumors with growth restricted by a specific T lymphocyte response, and therefore perioperative anti-inflammatory treatment has been proposed to reduce early metastatic recurrence in patients with breast cancer [143]. 

#### 3.2.2. Stress, Psychosocial Factors, and Physical Exercise

Regulation of stress levels and psychosocial factors not only improves patient quality of life but has also demonstrated physiological implications in the migratory activity of carcinoma cells as regards levels of norepinephrine, dopamine, and substance P, which depend on the activation state of the SNS [144,145]. In addition, exogenous and endogenous glucocorticoids such as cortisol lead to an increase of glucocorticoid receptors and favor metastatic spread in patients with breast cancer, while inducing an immunosuppressed state that favors tumor progression [11,12]. Furthermore, the sleep-wake cycle, regional brain activity, and behavior also modify psychoneuroimmune regulatory functions and are determining factors in the pathogenesis of the tumor [48,61,146].

Physical exercise is another way to regulate emotional state in people with cancer, and as we have already seen, to contribute to the remodeling of TME through the HPA axis. The physiopathogenic mechanisms involved are currently a matter of preferential attention for the medical and research community [147], which has led to its promotion from a purely preventive consideration to a first-level therapeutic tool, especially associated with immunotherapy [148], with important physiological effects on inflammation and immunity, as well as hormonal, metabolic and IMB status. The impact of exercise on cancer in terms of mortality, recurrence, and treatment-related adverse effects has been demonstrated [149], and its histopathological bases are being studied in depth as the basis for a crucial complementary therapeutic approach in cancer.

#### 3.2.3. Vitamin D3 and Vitamin D Receptor (VDR) 

Serum vitamin D3 concentration is associated with all-cause mortality [150] and with survival in the most common cancers such as colon, breast, or lung [151,152,153]. This hypothesis was also supported for ovarian cancer, pancreas, prostate, and invasive tumors considered all together in another study [154]. The mechanism of action relies on genomic regulation of expression levels of cell signaling and differentiation pathways key regulators. Part of the process involves activating intracellular signaling pathways that allow differentiation of myeloid cells [155], and specifically, the capacity of stromal reprogramming mediated by the vitamin D receptor (VDR), with the ability to modify the TME and its response to treatment [156] (Figure 3). In addition, vitamin D3 regulates the IMB and with this the TEM, as well as the host immune response [157]. 

#### 3.2.4. Metformin and Other Biguanides

Discovering drugs that can regulate metabolism, without cytotoxic effects but regulating TME and immunity while avoiding unwanted side effects and complications is particularly vital in tumors with insulin dependence, such as breast, prostate, ovary and endometrium cancers [158], especially as the available epidemiological data links metabolic status, overweight and obesity to cancer [159]. Since energy metabolism reprogramming is considered a new hallmark of cancer, drugs with metabolic effects acting on tissue remodeling have recently received attention [160]. Metformin and other biguanides have a marked impact on the TME and host immune response [139]. Metformin is an oral antidiabetic drug that inhibits mitochondrial complex I and oxidative phosphorylation increases CD8+ levels and modulates Treg cells in the TME, which clinically correlates with better outcomes in different tumors [160]. Interestingly, metformin not only acts at the hepatic level, mainly via the metabolism but also produces an indirect hypoglycemic effect through its impact on the IMB [161].

#### 3.2.5. Phytochemicals Derived from Natural Sources 

Various phytochemicals derived from natural sources, such as curcumin, ursolic acid, resveratrol, thymoquinone, and γ-tocotrienol, show different effects on the structure of TME, its function, and its metabolism, and remain to be studied in-depth as basic systems of stromal reprogramming and regulation of the TME [162].

Curcumin, the active component of turmeric, has been studied for its anti-inflammatory and anti-cancer effects. It suppresses the onset, progression, and metastasis of a variety of tumors in vitro and in vivo, and its effects are predominantly mediated through the negative regulation of various transcription factors, growth factors, inflammatory cytokines, protein kinases and especially their effect on the vasculature, fibroblasts, apoptosis, and metabolic regulation of TME [163]. Recently, it has been shown that alone or in combination with other substances, it affects the structure of the tumor vessels, normalizing the tumor vascular pattern and inhibiting tumor growth in an orthotopic nude mouse model of hepatocarcinoma [164]. Furthermore, curcumin shows a wide range of stromal effects, such as overcoming stromal protection of chronic lymphocytic leukemia B cells in vitro [165]. Moreover, it caused a decreased release of EMT-mediators in carcinoma-associated fibroblasts and reversal of EMT in tumor cells, which was associated with decreased invasion [166]. An in vitro inhibitory effect has been suggested for curcumin on lipogenic enzymes, and thus on cancer cell line progression, although further corroborative evidence from preclinical and clinical studies is required [167]. Its mechanism of action is not restricted to a single effect, but rather it acts through a broad range of functions and signaling pathways, as shown in human myeloid leukemia and embryonic kidney cells [168]. An advantage of using curcumin as a therapeutic agent is that there is no dose threshold linked to toxicity [169].

#### 3.2.6. Tensional Homeostasis

The emerging field of mechanobiology aims to study the mechanical properties associated with tissue morphogenesis, cell-cell or cell-matrix interactions, cellular migration, tumorigenesis, and progression of cancer [15]. As described in the section on biotensegrity and mechanotransduction, cells can sense the mechanical forces in their surroundings and modify their molecular behavior. ECM stiffness, shear stress, increased interstitial fluid pressure, and elevated interstitial fluid flow are all sensed and processed through mechanotransduction, eliciting a biological response [15]. Although growing evidence is pointing towards the importance of the immediate TME biophysics and its impingement on tumors, little is known about the systemic biophysical effects on cancer development [132,170,171]. However, if cells have the machinery to sense physical forces, such as mechanical load or traction forces, the different tissues (especially connective tissue), organs (particularly HPA axis and bone marrow), and systems (mainly the vascular system) in the human body serve as a platform to transmit systemic biophysical cues within the organism [172,173,174]. Based on distinctive physical features of tumor cells, acoustic-based approaches have been developed representing the significant potential for circulating tumor cell (CTC) sorting and diagnosis [175]. Recently, researchers have developed a platform that combines acoustics and microfluidics to isolate rare CTCs from peripheral blood in high throughput [176]. This achievement highlights how the exploitation of differential physical properties of cancer cells can lead to the development of diagnostic tools and prognosis. In addition, differential sorting of these cells has also demonstrated therapy response, in which stiff cells are more sensitive to chemotherapy than softer cells [177]. Further research is necessary to pinpoint the mechanisms by which cells react to external mechanical cues [171], to be able not only to establish diagnostic tools but also propose emerging alternative cancer therapies based on mechanotherapy.

## 4. Evaluation of New Biomarkers

The evaluation of new prognostic and predictive biomarkers focuses on multiple cellular subpopulations, molecules and metabolites of the TME that can be analyzed through precise tools: IHC, DNA/RNA/miRNA sequencing, flow cytometry, and mass flow cytometry. Besides genetic lesions, novel regulators of tumoral development are emerging, such as cell-cell and cell-ECM communication, HPA axis, microbiota, and mitochondrial dysfunction (Figure 4). In this context, the search for new biomarkers with clinical and therapeutic applications can be classified into four different groups: immune escape mechanisms, immune composition and activity in tumors, tumor-intrinsic factors, and host factors [178].

The evaluation of immune composition/activity and tumor/systemic biomarkers can be employed to predict the success of a therapeutic cycle, through analysis of the TME and the immune profile of the patient before starting treatment [179,180,181]. The different grade of the stroma [136] and the strength of the host immune response at both local and systemic levels all determine the efficacy of the immune antitumoral response [182]. Other parameters, which include the study of the intestinal microbiota and tumoral metabolism, are in process of being included in an integrated evaluation to identify, design, and apply anti-tumoral treatments tailored to each specific patient in the framework of personalized oncology.

Several hypotheses in cancer physiopathology assume that lymphoid cell activity is the principal mechanism involved in tumorigenesis inhibition, through cell immunity and secretion of several cytokines such as TNF (tumor necrosis factor) or IFN-γ, the complement system, opsonization, and others. This observation implies that processes such as inhibition of immunosuppressive cells (such as macrophages) or modulation of factors affecting the immune system (such as the microbiota), ultimately affect lymphoid cell activity, and support a relationship between tumor and immunity, such as the so-called “cancer immunogram” [179]. The most advanced therapies, such as immunotherapy, without a direct cytotoxic effect, reach response rates of 20–30% [183], results could be improved by developing immunotherapy with modified T-cells. On the other hand, adoptive immunotherapy has demonstrated outstanding results in a minority of hematological tumors (relapsed acute lymphoblastic leukemia and diffuse large B cell lymphoma) and the success of this approach seems to rely in the dominant role of CD19 as a target for ex-vivo technically modified lymphocytes. Although the focus of substantial research during the last few years results in solid tumors remain unfortunately poor. The elusive efficacy of adoptive immunotherapy in this scenario can probably be explained by the intrinsic heterogeneity that most solid tumors share with several different mutations, as expressed in different sections of the manuscript. Nonetheless, some progress has recently been made. Indeed, as presented in the latest online edition of ASCO 2020 (American Society of clinical oncology), adoptive T-cell therapy with ADP-A2M4 targeting MAGE-A4 has shown early activity in patients with different advanced solid tumors [184]. Therefore, the main limitation for adoptive immunotherapy lies in identifying the proper target in every tumor, to guide immune cell attack. Whether TME components should also be targets or not remains so far unknown. To overcome the limited patient response rate, severe side effects, and elevated costs of immunotherapy, it is crucial to discover novel biomarkers to predict which patients will respond. As a general rule, the best responders are patients with higher numbers of CD4+/CD8+lymphocytes, and in some cases, myeloid cells polarized to have a response similar to T helper lymphocytes [185]. Tumor subtypes with lower host immune response and with increased levels of the TGF-β, with a high content of M2 macrophages and an immunosuppressed TME tend to have the worst outcomes [186], while the subtypes rich in IFN-γ and with an inflammatory profile were found to be much more frequent, to share a Th1-like immune response and show more favorable prognosis [186]. Several biomarkers can be used to predict treatments based on the anti-cytotoxic T lymphocyte antigen-4 (CTLA-4) and against PD-1. While memory T-cell CD4+ and CD8+ have a fundamental role in both the CTLA-4 and PD-1 response, NK cells correlate with clinical response to anti-PD-1 treatment [187] (Table 1). Independently, PD-1-negative T cells can be rescued through co-stimulation with OX-40 or 4-1BB antagonists, which has shown that TIME subtype T3 tumors can be rewired to TIME subtype T2 [42].

## 5. Conclusions

The enormous plasticity that enables tumor cells to modify their phenotype and function [188] combined with intratumoral heterogeneity, make the TME a complex element of study and evaluation, decisive in the effectiveness of cancer treatment, especially immunotherapy [189]. Taken together, these observations invite a comprehensive reexamination of current cancer protocols and a shift towards new, more effective and safer therapeutic and follow-up horizons. The main unwanted effects of classical therapies are caused by altering the TME and inducing a pro-inflammatory response. Systemic changes derived from bone marrow, microbiota and CNS regulate the degree of cancer resistance at the macroscopic, microscopic and mesoscopic levels, and could be starting points in the trial of new therapeutic approaches [190]. Finding a “cure for cancer” is still beyond the reach of our current knowledge and technology. The available evidence suggests that the principal treatment modalities, including radiotherapy, chemotherapy, and surgical procedures, may induce an increase in circulating tumor cells, with the consequent heightened risk of distant metastases [191,192]. The clinical significance of circulating tumor cells (CTCs), circulating tumor DNA (ctDNA), and exosomes for diagnosis and prognosis of cancer should be underlined as should liquid biomarkers documenting the real-time monitoring of tumor evolution and therapeutic responses for personalized medicine. 

Host immune response composition and activity can be evaluated using predictive and prognostic tools such as the Immunoscore or TIME. The impact of immune cells such as B cells, natural killers, MDSC, macrophages, and T cells shows that tumors originating from different tissues are unique and possess a differentiated immune profile [131,193]. Due to considerable intratumoral heterogeneity, a PD-1/TILs positive or negative tumor can be mischaracterized when interpreting from a small biopsy sample; in this scenario, some PD-1 negative tumors may respond to anti-PD-1 therapy. In vitro cell-based assays are critical for probing the complex and dynamic nature of the interactions between the immune and cancer systems. Novel types of organ-on-chip models, based on a series of different human 3D-cell culture micro-chambers and immune cells, will allow us to understand mechanisms of resistance to immunotherapy and to evaluate in vitro strategies to overcome them [194].

The view of cancer as a metabolic disease points to sugar and fat consumption by the tumor, the importance of systemic support factors such as caloric restriction, entosis and autophagy, the role of the ketogenic diet, and mitochondrial metabolic therapy. PD-L1 can also regulate cancer cell metabolism through the mTOR pathway and extracellular glucose availability [195]. Metabolic TME remodeling could be achieved by exhausting tumor-promoting immune cells and promoting glycolysis in newly generated T cells, through a combination of inhibitory immune therapy against CTLA-4 (which diminish Treg cells) and metabolism inhibitors, with the consequent increase of effector T cells [126]. 

Rewiring different TME elements, including components of the nervous system, biotensegral structures at all body levels, and microbiota could constitute a starting point for novel therapies based on an extended oncology framework and an integral vision of cancer. Collectively, these observations invite us to rethink current oncological and pathological protocols and advance in our discovery of new biomarkers/tools to efficiently manage safe therapeutic follow-up.

## Figures and Tables

**Figure 1 cancers-12-01677-f001:**
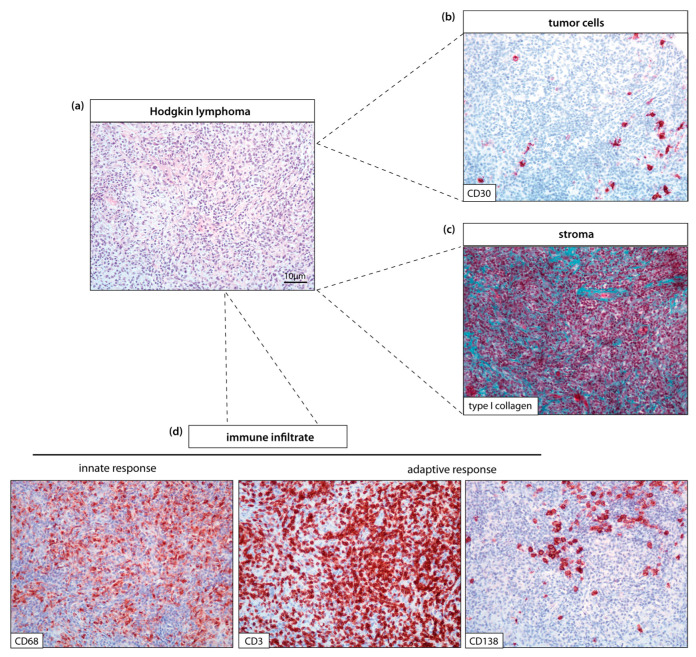
Hodgkin lymphoma is a clear example of a neoplasm where the components of the TME largely exceed the number of tumor cells. The differential staining of the histological sections corresponds to the same field of the tumor, showing overlapping layers that reveal a heterogeneous composition of the TME. (**a**) H&E staining of Hodgkin lymphoma. (**b**) little proportion of CD30 positive lymphoma tumoral cells. (**c**) Masson’s trichrome stain of abundant type I collagen (blue) and (**d**) representation of the host innate immune response via macrophage infiltrate (CD68) and the adaptive cellular response mediated by T cells (CD3) and B cells secreting antibodies against the tumor (CD138). Among others, this example constitutes the complex immune and stromal response that determines the histology, response to treatment, and tumor prognosis.

**Figure 2 cancers-12-01677-f002:**
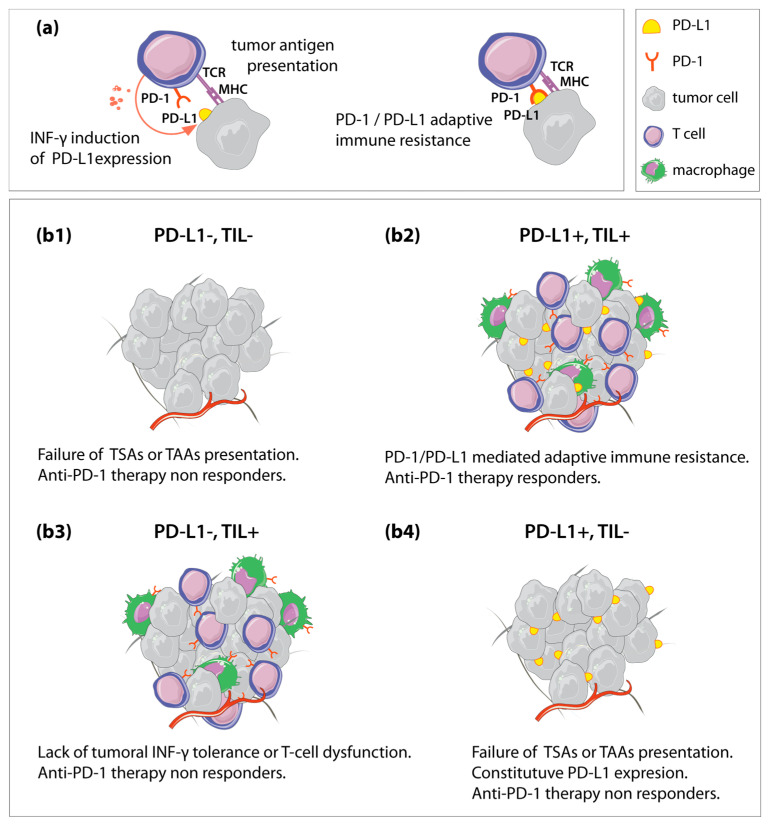
Schematic representation of TIME (Tumor Immune MicroEnvironment) classification. (**a**) The PD-1/PD-L1 pathway represents an adaptive immune resistance mechanism exerted by tumor cells in response to endogenous immune anti-tumor activity. Engagement of PD-L1 expressed on the tumor cells to PD-1 receptors on the activated T cells leads to inhibition of cytotoxic T cells. (**b1**–**b4**) Classification into 4 subtypes listed as TIME. (**b1**) PD-L1-, TIL− is classified into type 1 (T1). (**b2**) PD-L1+, TIL+, belongs to type 2 (T2). (**b3**) PD-L1−, TIL+ belongs to type 3 (T3) and (**b4**) PD-L1+, TIL−, classified as type 4 (T4) although its existence is under debate. MHC: major histocompatibility complex. TCR: T cell receptor. TAAs: tumor-associated antigens. TSAs: tumor-specific antigens. Legends at the top right.

**Figure 3 cancers-12-01677-f003:**
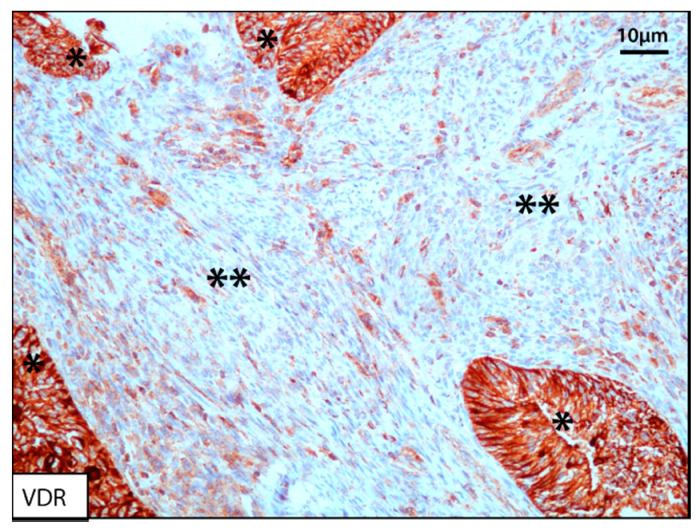
The reprogramming capacity of the tumor stroma is largely due both to the indirect effect of vitamin D on the microbiota and the presence of vitamin D receptor (VDR) on the components of the TME (**). Carcinoma cells (*). Image acquired at 20×.

**Figure 4 cancers-12-01677-f004:**
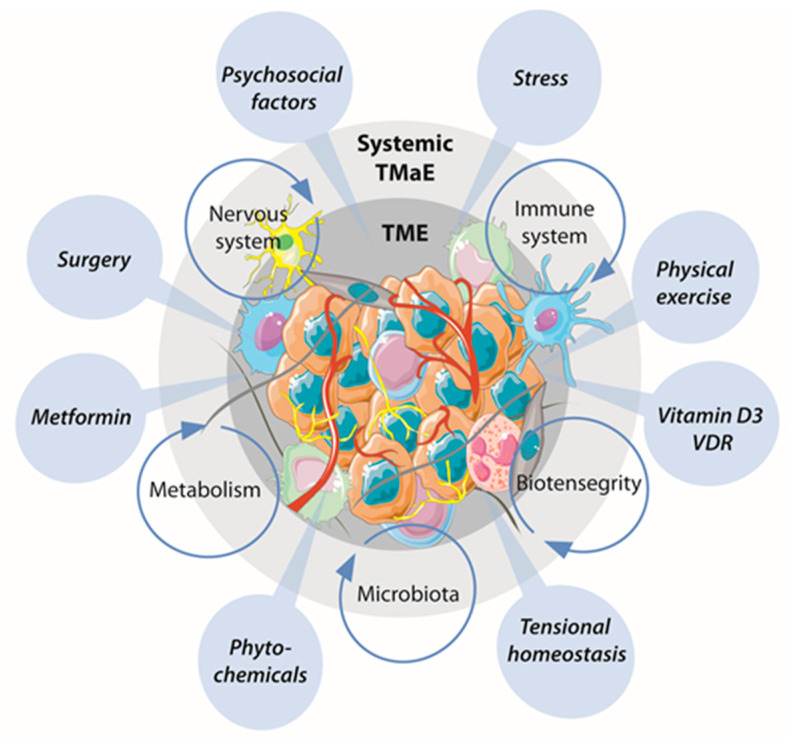
An integral vision of cancer biomarkers. TME elements are ultimately affected by systemic tumor macroenvironment (TMaE). The balance between them is a key determinant in tumor progression and aggressiveness. Therefore, analysis of this multi-level interaction could be beneficial for patient stratification and cancer therapy advancement and is also crucial for researchers in the field to improve current cancer models. Finally, emerging biomarkers need to be further explored and integrated to better understand the delicate information exchange occurring at the molecular/cellular/extracellular levels between the surrounding milieus. VDR: vitamin D3 receptor.

**Table 1 cancers-12-01677-t001:** Evaluation methods for emerging biomarkers.

Evaluation Groups	Parameters	Indicators	Detection	Method	References
TME	Inflammatory infiltrated cells	TAM	CD68, CD163	IHC	[79]
		TAN	CD15, CD32, CD35		[80]
		T helper	CD4		[81]
		Cytotoxic T cells	CD8		[82]
		Memory T cells	CD8, CD4, CTLA-4		[83]
		Tregs	FOXP3, CD4, CD25		[84]
		DC	CD141, CLEC9, CD11c		[85,86]
		NK	CD16, CD56, PD-L1, PD-L2		[87]
		B lymphocytes	CD20		[88]
	Stromal cells	CAF	αSAM, CD10, FSP1, AEBP1		[89,90]
		Schwann cells	S100, GFAP, p75NTR		[91]
	MSC	SC	Sox2, Oct4, CD133, Nestin, c-kit		[92]
	Fibers	Collagen type I	T. Masson, van Gieson	HC	[93,94]
		Collagen type III			[94]
		Elastic fibers	Orcein, Gomori, Snook, Wilder, Verhoeff		[94]
	Interstitial fluid	Proteoglycans	Alcian blue		[95]
		Fibronectin	Antifibronectin	IHC	[96]
		Laminin	Antilaminin		[97]
		Vitronectin	Antivitronectin		[98]
		Growth factor	TGF-β	AS	[99]
		Cytokines	Lymphokines	ELISA	[100]
		Proteases	Metalloproteinases		[100]
		Oxygen (ROS)	GSH/GSSG		[100]
	3D structure	Fibres and Cellular elements	Topology	Graph theory	[101,102]
	Mechanical forces	Focal adhesions	(F-actin, myosin II, α-actin, fascin)	IF	[103]
		Stress fibres			[103]
	Mechanotransduction	Mechano-actuated shuttling proteins	(β-catenin, zyxin)		[103]
		LINC complex	(SUN and nesprins)		[103]
Systemic factors	Glycolic index	↑metabolic index	18FDG	PET	[104]
	pH	Acidosis	Electrolytes serum concentration	Enzymatic	[105]
	Oxygen saturation	Hypoxia	↑HIF-1, ↑lactate	IHC	[106]
	Metabolism	Inflammatory response	↓VEGF	AS, ELISA	[107]
	Intestinal microbiota	Dysbiosis	↑*Bacteroids*	MALDI-TOF MS	[76,108,109,110]
			↑*Fusobacterium*	NGS	[76,108,109,110]
			↑*Porphyromonas*	16S rRNA	[76,108,109,110]
			↑*Enterobacter*		[76,108,109,110]
			↑*Cybrobacter*		[76,108,109,110]
	Nervous system	Deregulation	↑Norepinephrine,	HPLC	[62]
			↑Dopamine		[111]
			↑substance P		[112]
			↓β-endorphins		[113]

AS: absorption spectrometry; DC: dendritic cells; CAF: cancer-associated fibroblasts; 18FDG: fluorodeoxyglucose; GSH: glutathione; GSSG: glutathione disulfide; HC: histochemistry; HPLC: high-performance liquid chromatography; IF: immunofluorescence; IHC: immunohistochemistry; LINC: linker of nucleoskeleton and cytoskeleton; MALDI-TOF MS: matrix-assisted laser desorption/ionization time-of-flight mass spectrometry; MSC: mesenchymal stem cells; NGS: next-generation sequencing; NK: natural killer cells; PET: positron emission tomography; ROS: reactive oxygen species; 16S rRNA: ribosomal RNA 16S; SC: stem cells; TAM: tumor-associated macrophage; TAN: tumor-associated neutrophils; TME: tumor microenvironment; Tregs: regulatory T cells. Adapted from Noguera et at., 2019 [46].

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
