# Peer review of "Integrating the Tumor Microenvironment into Cancer Therapy"

_cancers, 2020, doi:10.3390/cancers12061677_

Round 1

Reviewer 1 Report

The review entitled “Integrating the Tumor Microenvironment into Cancer Therapy” submitted by Sabina Sanegre et al is very well written. The topic is very wide, and it was summarized in a comprehensive manner. The authors review and evaluate prognostic and predictive biomarkers of the tumor macroenvironment (TMaE), together with intratumoral heterogeneity which clearly affect tumor progression and patients responsiveness. The authors suggest that if we want to develop new therapeutics and better "control" the tumor, we need to consider not only the tumor but all the aspects of TMaE which include bone marrow, microbiota, CNS, among others. 

Author Response

We would like to thank the reviewer for the positive evaluation of the manuscript, her/his time and consideration.

Reviewer 2 Report

This paper review focuses on a new approach towards cancer therapy that focuses on the tumor microenvironment. The paper does a great job of explaining extrinsic stimuli with the capacity to reprogram the TME (section 3.2), but section 3.1 requires extensive repair of minor comments in order to make the paper easier to comprehend and make key findings more relevant.

Please address the following:

Minor comments:

  1. Introduction: The Cancer Genome Atlas (TCGA)

Please add a brief sentence explaining the major purpose of this atlas (e.g. identifying mutations in cancer)

  1. Introduction section: “Indeed, even the most clinically-promising drugs, such as tyrosine kinase inhibitors, represent a small advancement in comparison to the diversity of processes and pathway interactions regulated by these enzymes.”

Please add references based on the ability of these drugs to mutate and become resistant. A good source for lung cancer is PMID: 26667342 (e.g. lung cancer uses TKIs for EGFR mutations). This addition will help make your point that current therapeutic strategies have been unsuccessful because of 1) vastness of mutations and 2) resistance.

  1. Introduction section: …the idea that tumor microenvironment (TME)…

The definition of TME, and what encompasses is never truly defined, but rather the authors jump right into Section 2. Is the TME potentially reprogrammable?

  1. Emerging systems of TME reprogramming: Beyond the dynamic and progressive genetic alterations of cancer cells, other factors show multiple connections on the tumoral tissue with functional response capacity, such as TME molecules, hormones, cytokines and neurotransmitters, as well as the macroenvironment, where the intestinal microbiota and external factors ranging from stress to medication intake play a role.

3.1.1: However, most immunotherapies employed to date are administered systemically, leading to toxicity.

3.1.1 Sentences between reference 33 and 34. No references!!!

Table 1. States it is adapted from reference 36, could individual references be added? Please describe a few of these in text.

Please add appropriate references to back up these claims.

  1. Section 3.1.1: For information between references 33 and 34.

As currently constructed, this information is confusing/hard to follow. Some suggestions include:

Listing the 4 subtypes after the following sentence “The classification into 4 substypes….”

I currently only see two subtypes (I-E, I-I). Why is this? Perhaps say we will focus on two subtypes and why.

When saying “In comparison with the infiltrated-inflamed…” – We know nothing about I-I at this point, what is there to compare? Simply explain the general reasoning behind each subtype alongside their respective purpose and then compare and contrast the two!

  1. 3.1.3.: Recently, it has been shown that MYC and MCL1 amplification increase oxidative phosphorylation (OXPHOS) in breast cancer stem cells and promote chemotherapeutic resistance [60].

This statement is unnecessary (ref. 59 is enough), focus on components of the TME like glutamine levels (ref. 61), the focus of this paper.

  1. 3.1.3: End of the first paragraph. “[“ put instead of a period. Check the rest of the paper for any similar errors. Perhaps a reference was intended to be put here?
  2. What is the “intestinal microbiota (IMB)”?
  3. Please move the section 3.1.5. closer to sections 3.1.1 and 3.1.2 as they are interconnected (e.g. IMB, immunotherapy, NS).
  4. Section 4: Please include limitations for a few of the biomarkers. For instance, “These results could be improved with the development of immunotherapy with modified T-cells.” – How feasible is this?

Reviewer 3 Report

In this manuscript the authors provide a multifaceted overview of the TME and how this can be envisioned for cancer treatment. The Review is original, and the presentation is neat and well-written, I congratulate the authors for that.

I have some suggestions for improvement:

  • I think the Review would benefit of one (or two) additional illustrations; an example would be, for instance, a representation of the "4 subtypes listed as TIME". This would be nice for the reader to follow the text.
  • Authors could provide some histological images of actual tumor types that they mention, evidencing some of the features of the TME, like tumors highly infiltrated by specific immune cell types versus others with less infiltration; tumors with a lot of desmoplasia and stiffness; etc.
  • The authors should give more emphasis in the conclusions on exactly how we can therapeutically take advantage of this knowledge for treating cancer patients, for example by means of combination therapies, and also how we can deal with all this tumor and TME heterogeneity. How do they envision future studies to get to clinically relevant biomarkers or therapeutic targets?

Reviewer 4 Report

Authors reviewed comprehensive understanding of the tumor microenvironment. Authors carefully and deeply discussed the tumor microenvironment from a wide perspective. I agree that it is important for us to understand the features of the tumor microenvironment to develop an effective cancer therapy. However, there are some points to be assessed more.

At section 3.1.1. Remodeling TME to enhance antitumor immune activity, authors should conclude or suggest how we remodel TME to enhance immune responses against tumor before moving to next topic, the nervous system, even though authors mentioned it in the conclusion section.

Please rephrase the sentence in the first paragraph of 3.1.2 in page 4,”While there is increased insight into …..” because ‘function and significance’ is used twice in the sentence. It’s must be a typo or something.

Authors mentioned in the section 3.1.3 that prostaglandin E2 and interleukin 6 secreted by cancer can induce M2 macrophage and CD4+ Th1 cells can switch to activated M1. But I felt strange why authors discussed the helper T cell function. If there is some reason to touch helper T-cells there, please add the reason or something. Recently, metabolism of effector T-cells have been increasingly focused in the field of tumor immunity. So more discussions about that would be welcome.

Round 2

Reviewer 2 Report

I am satisfied with the revision.